# Lung Cancer Under Siege in Spain: Timeliness, Treatment, and Survival Before and After the COVID-19 Pandemic

**DOI:** 10.3390/cancers17162655

**Published:** 2025-08-14

**Authors:** Manuel Luis Blanco-Villar, José Expósito-Hernández, Eulalia Navarro-Moreno, Adrián Aparicio Mota, José María López Martín

**Affiliations:** 1Radiation Oncology Department, University Hospital Virgen de las Nieves, 18014 Granada, Spain; jose.exposito.sspa@juntadeandalucia.es; 2Instituto de Investigación Biosanitaria IBs, 18012 Granada, Spain; 3Tumor Registry of Torrecárdenas University Hospital, 04009 Almeria, Spain; eulalia.navarro.sspa@juntadeandalucia.es; 4Department of Preventive Medicine and Public Health, University Hospital Torrecárdenas, 04009 Almeria, Spain; 5Research Support Technician in the Field of Epidemiology, FIBAO and Biomedical Research Unit, Torrecárdenas University Hospital, 04009 Almería, Spain; aaparicio@fibao.es (A.A.M.); jmlopez@fibao.es (J.M.L.M.)

**Keywords:** lung cancer, COVID-19, diagnostic delays, wait times, survival outcomes, prognostic factors

## Abstract

The COVID-19 pandemic disrupted routine cancer care, raising concerns about delayed diagnoses and treatments, especially for lung cancer. In this study, we investigated the impact of the pandemic on the diagnosis, treatment timing, and survival of lung cancer patients at a large public hospital in Spain. We compared three consecutive years—before, during, and after the peak pandemic period—to see how care patterns evolved. Surprisingly, despite major disruptions in healthcare, we found that essential lung cancer care was largely preserved. Diagnosis and treatment times remained stable, and short-term survival even improved during the pandemic. These findings show that a strong, adaptive health system can maintain high-quality cancer care during times of crisis. Our results offer valuable insight into how hospitals can prepare for future challenges without compromising cancer outcomes.

## 1. Introduction

Lung cancer remains the leading cause of cancer-related mortality worldwide, accounting for approximately 18% of all cancer deaths [1]. Each year, more than two million new cases are diagnosed, and the five-year survival rate remains persistently low at approximately 10–20% [1]. Despite therapeutic advances, most patients are still diagnosed at advanced stages, and delays in diagnosis and treatment can further compromise prognosis [2].

The COVID-19 pandemic placed unprecedented pressure on healthcare systems worldwide, substantially disrupting oncology care delivery [3]. In particular, the reallocation of resources to COVID-19 units, overlapping respiratory symptoms between SARS-CoV-2 infection and lung cancer, and delays in patient presentation contributed to reductions in diagnostic activity and delays in treatment initiation [4,5]. Several national and international reports have documented marked declines—up to 30%—in new lung cancer diagnoses during the early phases of the pandemic, as well as procedural delays affecting imaging, biopsy, pathology, surgery, and systemic therapy [5,6,7,8]. In Spain, national registries and reports from the Spanish Thoracic Surgery Society (SECT) confirmed substantial reductions in thoracic surgery and interventional pulmonology procedures, along with delays in multidisciplinary decision-making and access to curative-intent therapies [5,8].

Timeliness in lung cancer care is critical, as even brief diagnostic or therapeutic delays may lead to stage migration and adversely affect prognosis [9]. In addition, interruptions in molecular testing or PD-L1 analysis have likely delayed the initiation of targeted therapies or immunotherapy in eligible patients, with potential clinical consequences [6]. In response to these challenges, several institutions implemented adaptive strategies to safeguard cancer care delivery [6]. These included prioritization of high-risk patients, the establishment of fast-track pathways, and streamlined diagnostic workflows [6]. However, responses varied significantly between centers and healthcare systems, making it difficult to evaluate the long-term effects of the pandemic on lung cancer outcomes in a standardized way [6,10]. This variability underscores the importance of institution-specific, real-world analyses, particularly as part of broader efforts to assess cancer care performance and resilience [11]. This study is part of a broader research initiative evaluating lung cancer management in Spain. It builds upon previous findings from a provincial registry study that revealed critical diagnostic and therapeutic delays even before the pandemic [11]. By comparing lung cancer diagnosis, treatment timelines, and outcomes across three consecutive years—before, during, and after the peak pandemic period—this analysis aims to quantify the real-world impact of COVID-19 on oncologic care and inform future preparedness strategies. We hypothesized that the pandemic would lead to a reduction in new lung cancer diagnoses, delays in diagnostic and treatment intervals, and a decrease in treatment initiation rates. These disruptions were expected to result in more advanced disease presentation and worse short-term survival compared to the pre-pandemic period.

## 2. Materials and Methods

### 2.1. Study Design

The study was conducted at a tertiary referral center that coordinates lung cancer care for the entire province of Almería, Spain. Although some patients were initially diagnosed at regional hospitals, all individuals included in this analysis were referred to the referral center for treatment planning and received their first oncologic intervention—whether radiotherapy, systemic therapy, or palliative care—at that facility. Surgical procedures, when indicated, were performed at the thoracic surgery unit of the University Hospital in Granada, the designated regional center for thoracic surgery. This centralized care model ensured consistency in clinical decision-making, data collection, and patient follow-up throughout the study period.

This retrospective cohort analysis aimed to evaluate changes in lung cancer care delivery associated with the COVID-19 pandemic. Diagnosis rates, treatment timelines, and clinical outcomes were compared across three defined periods: pre-pandemic (March 2019–March 2020), first pandemic year (March 2020–March 2021), and second pandemic year (March 2021–March 2022). The 2019 cohort, corresponding to the pre-pandemic year, served as the primary comparator for all analyses. Outcomes observed during the first and second years of the COVID-19 pandemic (2020 and 2021, respectively) were evaluated about this reference period. Ethical approval was granted by the hospital’s Institutional Review Board (IRB), and all patient data were fully anonymized to ensure confidentiality in compliance with the ethical standards outlined in the Declaration of Helsinki.

### 2.2. Inclusion and Exclusion Criteria

Eligible patients were adults (≥18 years old) with a radiological, clinical, or histopathological diagnosis of primary lung cancer who were first evaluated at our institution between March 2019 and March 2022. Patients were included if their electronic health records contained sufficient clinical information to determine diagnostic and treatment timelines, ECOG performance status, cancer stage, and treatment intent.

Patients were excluded if they met any of the following criteria: incomplete clinical records regarding diagnostic intervals or treatment initiation, or loss to follow-up prior to the start of treatment.

These criteria were consistent with those used in our prior analysis of the provincial lung cancer registry [11].

### 2.3. Study Population

A total of 594 patients with suspected primary lung cancer diagnosed between 1 March 2019, and 1 March 2022, were initially identified. After applying exclusion criteria, including incomplete diagnostic or treatment data, 530 patients were included in the final analysis.

The study population was stratified into three cohorts based on the period of diagnosis:

2019 (pre-pandemic): 1 March 2019–1 March 2020 (179 patients).

2020 (first pandemic year): 1 March 2020–1 March 2021 (145 patients, representing an 18.99% decrease compared to 2019).

2021 (second pandemic year): 1 March 2021–1 March 2022 (206 patients, reflecting a 42.07% increase compared to 2020).

A detailed flowchart of patient selection, including exclusions, is shown in Figure 1.

### 2.4. Data Collection and Assessment of Lung Cancer Care Timelines

Patient care involves multiple specialists, including diagnostic services, thoracic surgeons, medical oncologists, and radiation oncologists. The involvement of these specialists was systematically tracked, and the time intervals between consultations were recorded to assess potential delays and identify areas for improvement in care coordination. During pandemic peaks, emergency referrals were prioritized for rapid diagnostic work-up by internal hospital triage policies, while outpatient referrals—especially from primary care—were occasionally subject to delays.

Data were retrospectively extracted from electronic medical records, including demographic variables (age, sex, smoking history), clinical characteristics (ECOG performance status, cancer stage, histological subtype, and biomarker status), and oncologic treatment details such as the type of first definitive treatment (FDT) (surgery, chemotherapy, radiotherapy, or symptom-directed treatment). Additionally, relevant time points were collected, including symptom onset, referral dates, pathological diagnosis confirmation, multidisciplinary tumor board discussions, and treatment initiation dates. The full data collection process is illustrated in Figure 2.

Wait time intervals were derived from validated quality indicators, previously established in our study on diagnostic and treatment delays in lung cancer patients [11]. These indicators were adapted from international guidelines, including those issued by the British Thoracic Society [12] and the Lung Cancer Care Andalusia (LCCA) [13], ensuring a standardized and evidence-based approach to assessing lung cancer management.

This study evaluated the two principal diagnostic and treatment intervals: (A) referral to diagnosis, the time from specialist referral to obtaining a pathological diagnosis (Target: ≤30 days); and (B) diagnosis to first definitive treatment (FDT), representing the time from confirmed diagnosis to the initiation of surgery, chemotherapy, radiotherapy, or other oncologic treatments (target: ≤42 days).

To assess compliance with these time targets on patient outcomes, the study analyzed the proportion of patients who started treatment within ≤42 days of diagnosis and the impact on overall survival (OS) of receiving a diagnosis within ≤30 days from the first specialist consultation. The study also evaluated treatment trends over time, identifying variations in the first definitive treatments (FDTs) administered each year: 125 in 2019, 102 in 2020 (pandemic-related decline), and 151 in 2021 (post-pandemic recovery). These variations reflect pandemic-induced operational challenges and subsequent adaptations in the healthcare system.

All patients were staged according to the 8th Edition of the *TNM Classification for Lung Cancer* [14]. Non-small cell lung cancer (NSCLC) was categorized as early-stage (IA-IIB), locally advanced (IIIA-IIIC), or advanced/metastatic (IV), while small cell lung cancer (SCLC) was classified as limited or extensive-stage. The overall survival (OS) was defined as the time from the final pathology report to death from any cause or last follow-up. The data cutoff date was 1 January 2025, ensuring a minimum follow-up period of at least two years for most patients.

### 2.5. Statistical Analysis

Descriptive statistics were employed to summarize demographic data and wait times. Continuous variables were expressed as means with standard deviations or medians with interquartile ranges, depending on data distribution. Categorical variables were presented as proportions. Negative or inconsistent wait times (e.g., pathological diagnoses occurring after treatment initiation) were excluded from the analysis as outliers, ensuring accuracy in the assessment of timeliness.

Univariate Cox regression analysis was performed to examine associations between wait times and survival outcomes. Variables significant in univariate analysis were further evaluated using a multivariate Cox proportional hazards model, adjusting for potential confounders such as age, sex, ECOG performance status, cancer stage, histology, and treatment type. Statistical significance was defined as a *p*-value < 0.05. All analyses were conducted using R Statistical Software (v4.1.2) and SPSS version 26.

A sample size calculation was performed using stratified random sampling and Epidat 4.2 software. Based on a reference population of 530 lung cancer patients diagnosed across the three-year study period, the minimum required sample size was estimated at 224 cases, assuming a 95% confidence level, maximum variability (*p* = 0.5), and 5% precision. Stratification was conducted by year, with proportional allocation across strata. The final sample of 530 patients exceeded the calculated minimum and ensured sufficient power to detect relevant differences in survival across the three cohorts.

## 3. Results

### 3.1. Patient Characteristics

A total of 530 lung cancer patients were analyzed across three cohorts: pre-pandemic (2019, n = 179), first year of the pandemic (2020, n = 145), and second year of the pandemic (2021, n = 206). The demographic, clinical, and treatment characteristics of the patients are summarized in Table 1.

The median age of diagnosis increased, from 65 years in 2019 to 67 years in 2021 (*p* = 0.0368). Male patients accounted for the majority across all years, representing 77–82% of the population (*p* = 0.6280). Smoking history remained consistent, with over 90% of patients being current or former smokers in all cohorts (*p* = 0.2940). ECOG performance status showed a significant improvement during the pandemic, with 76.06% of patients in 2021 classified as ECOG 0-1 compared to 65.41% in 2019 (*p* = 0.0411).

Regarding histological subtypes, NSCLC remained predominant, accounting for approximately 80–85% of cases across cohorts. Within NSCLC, the proportion of adenocarcinoma decreased slightly from 66.17% in 2019 to 56.89% in 2021, while squamous cell carcinoma cases increased over the same period (*p* = 0.0601). Advanced/metastatic stage diagnoses decreased significantly during the pandemic, from 68.72% in 2019 to 56.31% in 2021 (*p* = 0.1087).

The proportion of patients receiving first definitive treatment (FDT) remained stable, ranging from 69.83% in 2019 to 73.30% in 2021 (*p* = 0.7203), while palliative treatment (PT) decreased slightly. Referrals from emergency departments declined from 58.66% to 50.59%, while inpatient consultations increased from 20.67% to 28.16%, with outpatient referrals from general practitioners (GPs) remaining stable (*p* = 0.4935).

### 3.2. Lung Cancer Diagnoses and Diagnostic Efficiency

The volume of new lung cancer diagnoses changed significantly during the pandemic, as shown in Table 2. The number of diagnoses decreased by 19% in 2020 compared to 2019, likely reflecting delays in healthcare access during the initial stages of the pandemic. However, diagnoses rebounded in 2021, showing a 42% increase compared to 2020. On average, 15 new diagnoses per month were made in 2019, dropping to 12 per month in 2020 before increasing to 17 per month in 2021.

### 3.3. Treatment Rates and Modalities

Of the 530 patients included in the final analysis, 378 (71.3%) received the first definitive treatment (FDT), while the remaining 152 (28.7%) did not undergo oncologic treatment. The most frequent reasons for not initiating FDT were poor ECOG performance status (≥2), extensive metastatic disease, or major comorbidities precluding active therapy. In selected cases, treatment was not initiated due to rapid clinical deterioration or explicit patient refusal. These patients were managed with the best supportive or symptom-directed care. The distribution of treatment modalities is summarized in Table 3. Surgery as the first definitive treatment showed a slight increase during the pandemic, from 12.29% in 2019 to 15.05% in 2021 (*p* = 0.6158). The use of definitive radiation therapy, including stereotactic body radiation therapy (SBRT), remained low, with a minor increase in 2021. Conversely, palliative radiation therapy decreased significantly, from 12.29% in 2019 to 4.37% in 2021 (*p* = 0.0048). The proportion of patients receiving systemic therapy, including chemotherapy and immunotherapy, remained stable across all years (*p* = 0.2133).

### 3.4. Timeliness of Diagnoses and Treatment

Pandemic-related disruptions significantly impacted diagnostic and treatment timelines. As shown in Table 4, median referral-to-diagnosis times improved notably, decreasing from 19 days in 2019 to 14 days in both 2020 and 2021 (*p* < 0.0001). However, delays persisted in specific areas. Pathology report wait times increased slightly, with the median time rising from 7 days in 2019 to 8 days in 2021 (*p* = 0.0001), while molecular testing times extended from 11 days in 2019 to 15 days in 2021 (*p* = 0.0226). Despite these delays, the interval from diagnosis to first definitive treatment (FDT) remained stable, with median times ranging between 33 and 35 days across cohorts (*p* = 0.5754).

As summarized in Table 5, compliance with national wait time standards varied across different intervals. The proportion of patients meeting the ≤30-day referral-to-diagnosis target increased slightly, from 80.98% in 2019 to 83.70% in 2021 (*p* = 0.6630). The percentage of patients receiving pathology reports within 7 days improved substantially, rising from 55.21% in 2019 to 73.37% in 2021 (*p* = 0.0002). However, adherence to the ≤14-day molecular test standard was inconsistent, peaking at 64.15% in 2020 before declining to 40.91% in 2021 (*p* = 0.0278). The proportion of patients starting FDT within 42 days remained stable across cohorts, with 65.25% in 2019 and 61.43% in 2021 (*p* = 0.6839).

### 3.5. Survival Outcomes

Survival trends during the pandemic are illustrated in Figure 3. The 1-year survival rate improved during the pandemic, rising from 37% (95% CI: 30–45%) in 2019 to 43% (95% CI: 36–52%) in 2020 and 2021. Two-year survival rates also increased, reaching 30% (95% CI: 24–37%) in 2021 compared to 22% (95% CI: 16–29%) in 2019. Furthermore, median survival showed a modest increase, progressing from 7.6 months in 2019 to 9.24 months in 2021. These findings suggest that short-term survival outcomes improved during the pandemic period, possibly reflecting advancements in clinical management and healthcare adaptation.

As shown in Table 6, multivariate analysis identified advanced-stage disease (HR = 6.62, *p* < 0.001) and poor ECOG performance status (ECOG > 2) (HR = 3.34, *p* < 0.001) as the strongest independent predictors of mortality. In contrast, the year of diagnosis and diagnostic delays (>30 days) were not associated with survival (HR = 0.98, *p* = 0.9; HR = 0.78, *p* = 0.2). Treatment modality did not significantly impact survival in the multivariate model. Palliative radiation therapy, which showed the highest mortality risk in the univariate analysis (HR = 17.0, *p* < 0.001), lost significance after adjustment (HR = 1.92, *p* = 0.2). These results confirm that tumor stage and functional status are the primary determinants of survival, with no independent effect observed for treatment type, diagnosis timing, or the pandemic period.

## 4. Discussion

The COVID-19 pandemic exerted unprecedented pressure on healthcare systems globally, with significant repercussions for oncology care. Lung cancer (LC), due to its high lethality and frequent late-stage presentation, was especially vulnerable to disruptions in diagnostic and therapeutic pathways. In this real-world, three-year cohort study from a Spanish tertiary hospital, we provide an in-depth analysis of the pandemic’s impact on LC care, including diagnostic volumes, treatment patterns, wait times, and survival outcomes. Our findings are contextualized within the most recent international evidence.

Our center observed a marked reduction in new lung cancer diagnoses in 2020 (−18.99%; from 179 to 145 cases), followed by a notable rebound in 2021 (+42.07%; 206 cases), a pattern consistent with international reports [15,16,17]. The magnitude of this rebound likely reflects not only the reactivation of routine diagnostic activity but also the accumulation of delayed or deferred cases from the previous year. However, comparative analysis of clinical stage, ECOG performance status, and treatment rates revealed no major differences between 2021 cases and those from other periods, suggesting that these “catch-up” cases were not clinically distinct. This diagnostic collapse and subsequent recovery mirrored findings from Canada (−34.7% in 2020, +97% in 2021) [15], as well as evidence from other countries that noted increases in metastatic presentations during later pandemic phases [18]. The overlap between COVID-19 and lung cancer symptoms, patient hesitancy, and the reallocation of diagnostic services contributed to these trends [19,20]. Our experience reinforces global observations that the pandemic significantly disrupted routine cancer care.

Importantly, although the absolute number of patients treated in our center followed diagnostic trends (FDT cases: 125 in 2019, 102 in 2020, 151 in 2021), the proportion of diagnosed patients receiving first definitive treatment (FDT) remained stable throughout the study period (69.83% in 2019, 70.34% in 2020, and 73.30% in 2021; *p* = 0.7203). This is in sharp contrast with the Canadian experience, where there was a significant reduction specifically in lung cancer surgeries (up to −64% during the pandemic), whereas our institution managed to maintain stable access to the first definitive treatment, regardless of modality [15,21]. These data underscore our institution’s ability to sustain treatment pathways despite fluctuating diagnosis rates.

Regarding disease stage at presentation, we did not observe the anticipated shift toward more advanced stages. The proportion of advanced-stage diagnoses decreased from 68.72% in 2019 to 56.31% in 2021, with early-stage cases slightly increasing (from 15.64% to 17.96%; *p* = 0.1087). This diverges from Japanese data reporting a significant increase in advanced-stage presentations [22], and from predictive modeling that estimated a 76% probability of stage shift after six months of diagnostic delay [23]. The preservation of early-stage diagnosis in our center may be attributed to streamlined diagnostic processes and robust pre-pandemic investments in care coordination [11], as well as the high proportion of patients initially assessed via emergency services during the pandemic period.

A notable finding was the significant improvement in diagnostic intervals. Median referral-to-diagnosis time was reduced from 19 days (IQR: 14–29) in 2019 to 14 days (IQR: 11–22) in both 2020 and 2021 (*p* < 0.0001), and compliance with the ≤30-day benchmark improved from 80.98% to 83.70%. When focusing on pathology turnaround, the median time from biopsy to definitive pathology report was 7 days in 2019 and decreased to 6 days in both 2020 and 2021 (*p* = 0.0001), with the percentage of patients receiving the report within 7 days improving from 55.2% to 73.4%. This diagnostic stability may in part be explained by the high proportion of emergency referrals, which were prioritized throughout the pandemic and processed via fast-track protocols. In contrast, elective referrals experienced variable delays depending on system capacity. However, molecular testing turnaround was adversely affected: median time increased from 11 to 15 days (*p* = 0.0226), and compliance with the ≤14-day target fell from 44.59% to 40.91% (*p* = 0.0278). These challenges are in line with other studies identifying molecular diagnostics as a critical bottleneck in oncologic care during COVID-19 [24,25,26].

Notably, diagnosis-to-treatment intervals in our cohort remained stable throughout the pandemic: the median time from diagnosis to initiation of definitive treatment was 33 days (IQR: 22–44) in 2019, 35 days (IQR: 22–46) in 2020, and 35.5 days (IQR: 24–50) in 2021 (*p* = 0.5754), with over 70% of patients consistently starting treatment within 42 days of diagnosis. This performance contrasts with data from the UK, where only 23% of patients initiated definitive treatment within the 62-day standard interval (from urgent referral to first treatment) during the pandemic [27], highlighting our institution’s ability to buffer upstream delays and preserve timely care despite global disruptions.

Access to curative treatments, particularly surgery and chemoradiotherapy, was preserved throughout the study period. The proportion of patients receiving such therapies increased modestly from 69.83% in 2019 to 73.30% in 2021, reflecting institutional prioritization during times of healthcare strain. Additionally, hospital avoidance behavior may have contributed to a reduced uptake of palliative services. Surgical intervention rates remained stable (12.29% in 2019, 15.86% in 2020, and 15.05% in 2021; *p* = 0.6158), as did rates of chemoradiotherapy and systemic therapy. Palliative radiotherapy, however, declined significantly (from 12.29% in 2019 to 4.37% in 2021; *p* = 0.0048), likely reflecting prioritization of high-value curative interventions during periods of maximum healthcare strain. These findings contrast with substantial declines in surgical access reported in Canada and Japan [15,18,22], but are aligned with recommendations for triage-based prioritization of curative care [19,20,28].

Contrary to initial concerns, we observed no negative impact on short-term survival. One-year survival improved from 37% (95% CI: 30–45%) in 2019 to 43% in 2020–2021, and two-year survival from 22% to 30%, with median survival rising from 7.6 to 9.24 months. Multivariate analysis identified advanced stage (HR = 6.62; *p* < 0.001) and ECOG ≥ 2 (HR = 3.34; *p* < 0.001) as the only independent predictors of mortality, while delays in diagnosis (>30 days; *p* = 0.2) and treatment (>42 days; *p* = 0.5) were not significant factors. These results support findings from Agulnik et al., who concluded that disease burden, particularly M1c status, rather than care delays, was the main determinant of mortality during the pandemic [18].

Time toxicity—the proportion of days patients spend in active contact with healthcare—is an emerging concern, particularly in the post-pandemic context. Johnson et al. recently reported that patients with advanced cancer spend up to 22% of their remaining time in healthcare settings, even during the pandemic [29]. Although this metric was not formally assessed in our study, our streamlined referral and treatment processes likely mitigated unnecessary contact. As highlighted by Bian et al. and Bouza et al., fragmentation or delay in care may exacerbate time toxicity, particularly among vulnerable populations [24,25]. Incorporating this parameter into future quality metrics would enable a more holistic evaluation of system performance. Broader metrics including screening delays have also been proposed [30].

This study has several limitations. First, its retrospective single-center design may limit generalizability and introduce selection bias, particularly when comparing cohorts across different phases of the pandemic. Second, although power was not recalculated post hoc, our initial sample size estimation confirmed that the final cohort exceeded the minimum required to detect relevant survival differences with acceptable precision. Third, the absence of cause-specific mortality data precluded the use of a competing risks model, which may have led to a modest overestimation of cancer-specific survival, especially during COVID-19 peaks. Fourth, comorbidity data, such as COPD or cardiovascular disease, were not consistently available and were therefore excluded from the multivariate model. While ECOG performance status served as a surrogate for overall patient condition, it may not fully reflect the impact of underlying comorbidities. Finally, we cannot exclude the possibility of lead-time bias, whereby earlier diagnosis—without changes in disease biology—could partially account for the observed improvement in short-term survival.

The key strengths of this study include the robust sample size of 530 well-characterized patients, the comprehensive three-year analysis encompassing distinct phases of the pandemic, and the use of real-world data from a Spanish tertiary hospital. By integrating diagnostic timelines, treatment patterns, and survival outcomes, the study offers a holistic assessment of lung cancer care during a period of exceptional healthcare disruption. Furthermore, comparison with international benchmarks enhances the relevance and generalizability of our findings. The consistency of clinical pathways, the stability of care delivery metrics, and the focus on institutional resilience provide additional depth to the evaluation, making the results highly informative for national preparedness planning.

Future studies should explore the long-term oncologic impact of pandemic-related disruptions, including disease recurrence, stage migration, and patient-reported outcomes. Additionally, further prospective multicenter research is needed to validate the protective effect of specific institutional strategies and to determine whether they remain effective in post-pandemic healthcare settings. It would also be of interest to assess whether similar patterns of care preservation and outcomes were observed in other tumor types, as differences in disease biology and treatment pathways may have led to variable impacts across malignancies.

## 5. Conclusions

In summary, although the COVID-19 pandemic caused significant disruptions to healthcare delivery, its overall impact on lung cancer outcomes in our institution was modest, thanks to the rapid adaptation and resilience of our care pathways. The findings of this study reaffirm that early stage at diagnosis and good performance status (ECOG 0-1) remain the most decisive prognostic factors, far outweighing the effect of diagnostic or treatment delays. Moving forward, future efforts should focus on strengthening early detection strategies and ensuring that efficient, patient-centered pathways remain a priority—even in times of crisis. Continuous evaluation of system bottlenecks, particularly in molecular diagnostics, will be essential to further minimize potential vulnerabilities in oncological care. These findings provide a real-world benchmark for the Spanish healthcare system and may help guide national preparedness strategies for future crises.

## Figures and Tables

**Figure 1 cancers-17-02655-f001:**
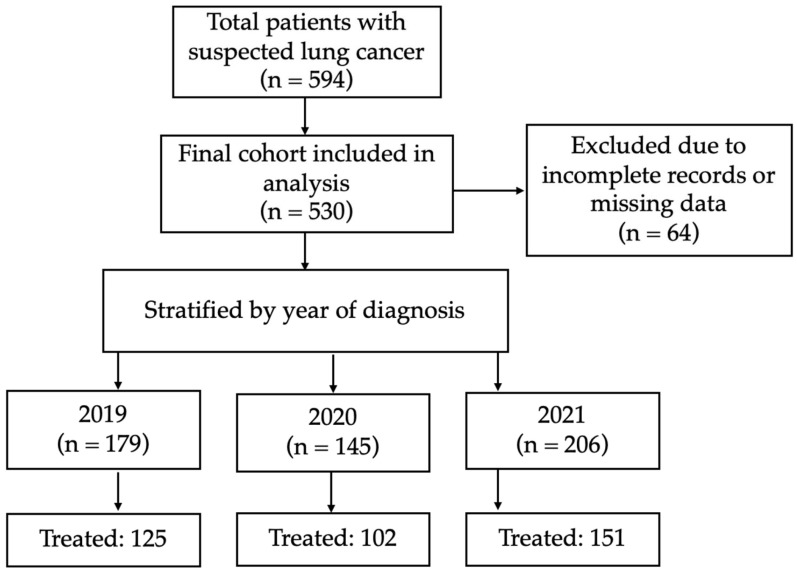
Patient flowchart. A total of 594 patients with suspected primary lung cancer were identified across three consecutive annual cohorts: pre-pandemic (2019, n = 179), first pandemic year (2020, n = 145), and second pandemic year (2021, n = 206). After applying exclusion criteria (e.g., incomplete clinical data), 530 patients were included in the final analysis. Among them, 378 received their first definitive treatment (FDT): 125 in 2019, 102 in 2020 (−19%), and 151 in 2021 (+48% vs. 2020). The figure shows relative year-on-year changes in diagnosis and treatment volumes: an 18.9% drop in new diagnoses from 2019 to 2020, followed by a 42% increase from 2020 to 2021.

**Figure 2 cancers-17-02655-f002:**
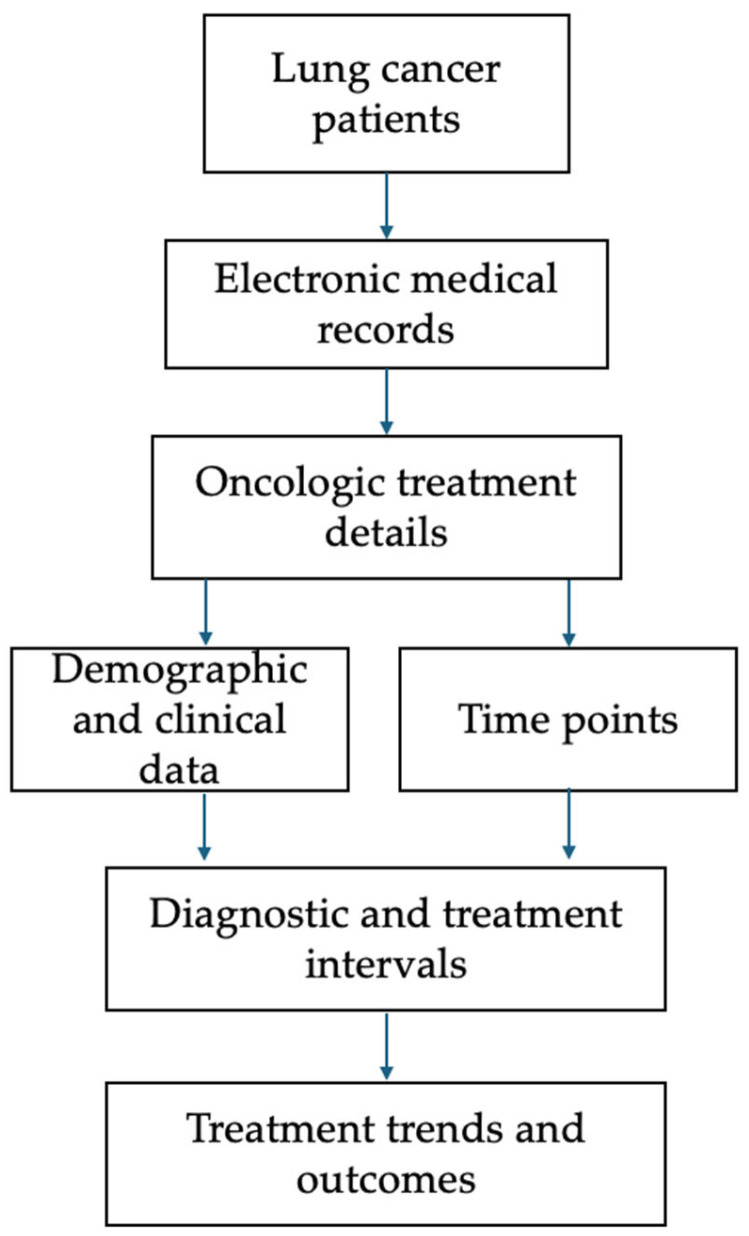
Flowchart of patient data collection and analysis process. The diagram summarizes the data collection pathway used in this study. Clinical, demographic, and oncologic variables were extracted from electronic medical records. Diagnostic time points and treatment details were combined to calculate standardized diagnostic and therapeutic intervals, enabling the analysis of care timeliness, treatment trends, and survival outcomes.

**Figure 3 cancers-17-02655-f003:**
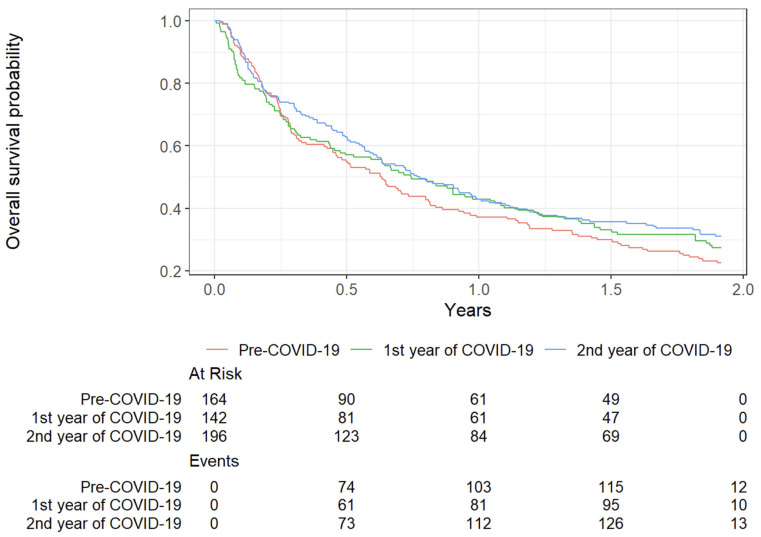
Overall survival curves in lung cancer patients: a comparison between the pre-COVID-19 period and the first two years of the pandemic. Kaplan–Meier survival curves showing overall survival probabilities in lung cancer patients across three periods: pre-COVID-19 (red), the first year of the COVID-19 pandemic (green), and the second year of the pandemic (blue). The “At Risk” table indicates the number of patients at risk at each time point, while the “Events” table shows cumulative deaths. Survival at 1 and 2 years remained stable or improved slightly during the pandemic periods compared to the pre-COVID-19 cohort, suggesting no significant negative impact on short-term outcomes.

**Table 1 cancers-17-02655-t001:** Patients characteristics.

Characteristics		2019 n = 179	2020 n = 145	2021 n = 206	*p*-Value
**Median** **age (years)**		65	64	67	0.0368
**Sex [n (%)]**	Male	146 (81.56%)	114 (78.62%)	160 (77.67%)	NS
	Female	33 (18.44%)	31 (21.38%)	46 (22.33%)	
**Smoking history** **[n (%)]**	Former/current smoker	168 (94.92%)	130 (90.91%)	187 (91.22%)	NS
	Non-smoker	9 (5.08%)	13 (9.09%)	18 (8.78%)	
**ECOG performance status [n (%)]**	0–1	104 (65.41%	96 (78.80%)	143 (76.06%)	0.0411
	>1	55 (34.58%)	29 (23.20%)	45 (23.94%)	
**Subtype/Histology** **[n (%)]**	NSCLC	135 (79.88%)	116 (85.29%)	169 (84.08%)	NS
	SCLC	34 (20.12%)	20 (14.71%)	32 (15.92%)	
	Unknown	10 (5.59%)	9 (6.21%)		
**NSCLC** **Subtype (n/%)**	Adenocarcinoma	88 (66.17%)	63 (55.26%)	95 (56.89%)	NS
	Squamous	33 (24.81%)	39 (34.21%)	64 (38.32%)	
	Others *	12 (9.02%)	12 (10.53%)	8 (4.79%)	
**Biomarkers**	EGFR Mutation				NS
	Positive Negative	9 (12%) 66 (88%)	6 (10%) 54 (90%)	6 (7.89%) 70 (92.1%)	
	ALK translocation				NS
	Positive Negative	0 (0%) 72 (100%)	2 (3.4%) 58 (96.6%)	2 (2.7%) 71 (97.3%)	
	PD-L1 expression				NS
	<1% 1–49% >50%	39 (40.2%) 28 (28.8%) 30 (31%)	50 (54.4%) 21 (22.8%) 21 (22.8%)	51 (48.5%) 28 (26.7%) 26 (24.8%)	
**Cancer Stage [n (%)]**	Early stage ^1^	28 (15.64%)	23 (15.86%)	37 (17.96%)	NS
	Locorregional ^2^	28 (15.64%)	34 (23.45%)	53 (25.73%)	
	Advanced/metastatic stage ^3^	123 (68.72%)	88 (60.69%)	116 (56.31%)	
**Treatment type** **[n (%)]**	FDT ^4^	125 (69.83%)	102 (70.34%)	151 (73.30%)	NS
	PT ^5^	54 (30.17%)	43 (29.66%)	55 (26.70%)	
**First consultation with pulmonary specialist [n (%)]**	Outpatient referral from GP	37 (20.67%)	30 (20.69%)	44 (21.36%)	NS
	Inpatient consult	37 (20.67%)	37 (25.52%)	58 (28.16%)	
	Referral from emergency department	105 (58.66%)	78 (53.79%)	104 (50.59%)	

Note: NS = non-significant (*p* ≥ 0.05). Statistical comparisons were performed using chi-square tests for categorical variables and Kruskal–Wallis tests for continuous variables. Statistical analyses were conducted using R (v4.1.2) and SPSS (version 26). A *p*-value < 0.05 was considered statistically significant. ECOG = eastern cooperative oncology group; NSCLC = non-small cell lung cancer; SCLC = small cell lung cancer; EGFR = epidermal growth factor receptor; ALK = anaplastic lymphoma kinase; PD-L1 = programmed death-ligand 1. ^1^ T1-3N0-1M0; ^2^ T1-4N2-3M0 + Limited SCLC; ^3^ TanyNanyM1 + Extensive SCLC. FDT ^4^ = first definitive treatment; PT ^5^ = Palliative treatment. * Includes: adenosquamous carcinoma, sarcomatoid carcinoma, carcinoid, large cell carcinoma, neuroendocrine tumor NOS, NSCLC NOS, undifferentiated carcinoma, and mixed carcinoma.

**Table 2 cancers-17-02655-t002:** Lung cancer diagnoses in the three cohorts.

Variables	2019	2020	2021
Number of new lung cancer diagnoses	179	145	206
Average per month	15	12	17
Change vs. previous year	−	−19%	+42%

Note: 2020 and 2021 are compared to the immediately preceding year. All figures are descriptive.

**Table 3 cancers-17-02655-t003:** Type of first definitive treatment (FTD).

Type of FTD	Total	2019 n = 125	2020 n = 102	2021 n = 151	*p*-Value
**Surgery (n/%)**	76 (14.34%)	22 (12.29%)	23 (15.86%)	31 (15.05%)	NS
**Definitive radiation (n/%)**	22 (4.15%)	7 (3.91%)	4 (2.76%)	11 (5.34%)	NS
Conventional	12 (2.26%)	4 (2.23%)	0 (0.00%)	8 (3.88%)	
SBRT	10 (1.89%)	3 (1.68%)	4 (2.76%)	3 (1.46%)	
**Definitive chemoradiation (n/%)**	79 (14.90%)	20 (11.17%)	22 (15.18%)	37 (17.96%)	NS
Concurrent chemoradiation	44 (8.30%)	13 (7.26%)	14 (9.66%)	17 (8.25%)	
Sequential chemoradiation	35 (6.60%)	7 (3.91%)	8 (5.52%)	20 (9.71%)	
**Palliative radiation (n/%)**	38 (7.17%)	22 (12.29%)	7 (4.83%)	9 (4.37%)	0.0048
**Systemic (n/%)**	163 (30.75%)	54 (30.17%)	46 (31.72%)	63 (30.58%)	NS
Standard systemic chemotherapy	111 (20.94%)	42 (23.46%)	32 (22.07%)	37 (17.96%)	
Immunotherapy ± Chemotherapy	41 (7.74%)	9 (5.03%)	9 (6.21%)	23 (11.17%)	
Targeted therapy	11 (2.08%)	3 (1.68%)	5 (3.45%)	3 (1.46%)	
**Symptom-directed (palliative care)**	152 (28.68%)	54 (30.17%)	43 (29.66%)	55 (26.70%)	NS

Note: NS = non-significant (*p* ≥ 0.05). Statistical comparisons were performed using chi-square tests for categorical variables and Kruskal–Wallis tests for continuous variables. Statistical analyses were conducted using R (v4.1.2) and SPSS (version 26). A *p*-value < 0.05 was considered statistically significant.

**Table 4 cancers-17-02655-t004:** Median wait times (days ^1^) before and during the two years of the COVID-19 pandemic.

Interval	2019	2020	2021	*p*-Value
**Referral → LC specialist** (≤7 days)	2 (1.0; 7.0)	1 (1.0; 5.0)	1 (1.0; 6.0)	NS
**Referral → Diagnosis** (≤30 days)	19 (14.0; 29.0)	14 (11.0; 22.0)	14 (11.0; 22.2)	<0.0001
**Pathology report** (≤7 days)	7 (6.0; 9.5)	6 (6.0; 8.0)	6 (6.0; 8.0)	0.0001
**Molecular results** (≤14 days)	11 (8.0; 18.0)	13 (8.0; 21.0)	15 (10.5; 22.0)	0.0226
**Diagnosis → FDT ^2^** (≤42 days)	33 (20.0; 50.5)	34.5 (23.25; 49.0)	35.5 (23.0; 54.0)	NS

Note: NS = non-significant (*p* ≥ 0.05). Statistical comparisons were performed using chi-square tests for categorical variables and Kruskal–Wallis tests for continuous variables. Statistical analyses were conducted using R (v4.1.2) and SPSS (version 26). A *p*-value < 0.05 was considered statistically significant. ^1^ Interquartile range, ^2^ first definitive treatment.

**Table 5 cancers-17-02655-t005:** The proportion of patients meeting the wait time standards.

Interval (Days)	2019	2020	2021	*p*-Value
**Referral → LC specialist** (≤7 days)	77.69%	81.44%	80.54%	0.7518
**Referral → Diagnosis** (≤30 days)	80.98%	84.80%	83.70%	0.6630
**Pathology report** (≤7 days)	55.21%	74.40%	73.37%	0.0002
**Molecular results** (≤14 days)	44.59%	64.15%	40.91%	0.0278
**Diagnosis → Treatment** (≤42 days)	65.25%	66.67%	61.43%	0.6839

Note: NS = non-significant (*p* ≥ 0.05). Statistical comparisons were performed using chi-square tests for categorical variables and Kruskal–Wallis tests for continuous variables. Statistical analyses were conducted using R (v4.1.2) and SPSS (version 26). A *p*-value < 0.05 was considered statistically significant.

**Table 6 cancers-17-02655-t006:** Overall survival model (n = 530).

Variables	Univariate Analysis		Multivariate Analysis	
	HR (95% CI)	*p*-Value	HR (95% CI)	*p*-Value
Age (vs ≤ 70 years)	1.15 (0.86, 1.53)	NS	1.23 (0.89–1.70)	NS
Female sex (vs. male)	0.64 (0.47, 0.96)	0.027	0.84 (0.57–1.23)	NS
Ever-smoker (vs. never)	1.84 (1.21–2.86)	0.008	1.79 (0.90–3.57)	NS
ECOG performance status (vs 0–1) ≥2	3.97 (2.87, 5.50)	<0.001	3.34 (2.28, 4.90)	<0.001
Histology (vs. SCLC)	0.51 (0.07, 3.68)	<0.001	0.70 (0.49, 0.98)	0.038
Stage (vs. Early stage-NSLC) Locally-Advanced + Limited Advanced + Extended	3.87 (2.21, 6.76) 9.29 (5.53, 15.6)	<0.001 <0.001	3.28 (1.26, 8.52) 6.62 (2.53, 17.3)	0.015 <0.001
First treatment (vs. Surgery) Definitive radiation Definitive chemoradiation Palliative Radiation Systemic (chemo and/or immunotherapy)	2.08 (0.95, 4.55) 2.88 (1.64, 5.08) 17 (9.34, 30.93) 7.12 (4.28, 11.8)	NS <0.001 <0.001 <0.001	0.89 (0.36, 2.21) 0.72 (0.30, 1.74) 1.92 (0.77, 4.75) 0.94 (0.40, 2.19)	NS NS NS NS
Year of diagnosis (vs. Pre-COVID-19) 1st year of COVID-19 2nd year of COVID-19	0.74 (0.53, 1.04) 0.78 (0.58, 1.06)	NS NS	0.98 (0.68, 1.41) 1.18 (0.84, 1.64)	NS NS
Wait times 1st appointment → Diagnosis ≥30 days Diagnosis → 1st treatment ≥42 days	0.74 (0.51, 1.06) 0.82 (0.61, 1.16)	NS NS	0.78 (0.54, 1.14) 0.90 (0.65, 1.24)	NS NS

Note: NS = non-significant (*p* ≥ 0.05). Statistical comparisons were performed using chi-square tests for categorical variables and Kruskal–Wallis tests for continuous variables. Statistical analyses were conducted using R (v4.1.2) and SPSS (version 26). A *p*-value < 0.05 was considered statistically significant.

## Data Availability

The data presented in this study are available on request from the corresponding author. The data are not publicly available due to institutional data protection policies and ethical restrictions related to patient confidentiality.

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
