# Peer review of "Lung Cancer Under Siege in Spain: Timeliness, Treatment, and Survival Before and After the COVID-19 Pandemic"

_cancers, 2025, doi:10.3390/cancers17162655_

Round 1

Reviewer 1 Report

Comments and Suggestions for Authors

Major Revision Recommendation:

While this study provides valuable insights into lung cancer care during COVID-19, several limitations warrant clarification. The retrospective design may introduce selection bias, particularly in comparing pre-pandemic and pandemic cohorts. The improved survival during the pandemic contradicts global trends and requires further validation, including adjustment for potential confounders like stage migration or treatment modifications. The molecular testing delays (11 to 15 days) and their clinical impact should be explored in depth. Additionally, the manuscript would benefit from a discussion of institutional strategies that preserved timeliness, as these could serve as a model for crisis preparedness. Clarifying these aspects would strengthen the study’s conclusions.

  1. The study lacks power calculation. With 18-42% annual variations in case volume, was the sample size adequate to detect clinically meaningful survival differences (e.g., 5% absolute change in 1-year survival)?
  2. How were emergency vs. outpatient referrals triaged differently during pandemic peaks? This could explain the stable diagnostic intervals despite system stress.
  3. For time-to-event analysis, why wasn't a competing risks model used given high mortality rates? COVID-19 deaths may have artificially inflated cancer survival estimates.
  4. The multivariate model doesn't adjust for comorbidities (e.g., COPD) that influence both treatment eligibility and survival. How might this affect the null finding for delay impacts?
  5. Molecular testing delays (11→15 days) are clinically significant for targeted therapy eligibility. Was there differential survival impact in biomarker-positive subgroups?
  6. The 42% diagnostic rebound in 2021 exceeds most published recovery rates (typically 10-25%). Could this reflect pent-up demand from 2020, and were "catch-up" cases clinically distinct?
  7. How do you reconcile stable treatment intervals with documented pathology/molecular delays? Were treatment protocols modified (e.g., empiric chemo before biomarker results)?
  8. The improved survival despite pandemic stresses contradicts modeling studies predicting 5-15% excess mortality. Could lead-time bias from earlier diagnosis explain this?
  9. Why did palliative radiotherapy decline sharply (12.3%→4.4%, p=0.0048)? This may reflect resource reallocation rather than clinical factors.

Comments on the Quality of English Language

Major Revision Recommendation:

Author Response

For research article

Response to Reviewer 1 Comments

1. Summary

We sincerely thank Reviewer 1 for the thoughtful and constructive comments, which have significantly contributed to improving the clarity, methodological rigor, and scientific relevance of our manuscript. We have carefully addressed each of the points raised and made the corresponding revisions in the updated version, which are clearly highlighted using track changes. Our responses below provide a point-by-point explanation and justification for the changes implemented in the manuscript.

2. Questions for General Evaluation

Reviewer’s Evaluation

Response and Revisions

Does the introduction provide sufficient background and include all relevant references?

Can be improved

We expanded the background with updated references and clarified the study rationale.

Is the research design appropriate?

Can be improved

We acknowledge the retrospective design has limitations. These are now discussed explicitly.

Are the methods adequately described?

Must be improved

Methods were revised for clarity.

Are the results clearly presented?

Can be improved

Tables were revised for clarity; NS was added for non-significant p-values. Table 2 was simplified and clarified.

Are the conclusions supported by the results?

Can be improved

We revised the discussion to acknowledge potential confounders and lead-time bias.

Are all figures and tables clear and well-presented?

Must be improved

Figures and Tables 1–6 were reformatted, footnotes were added, and statistical methods clarified below tables.

3. Point-by-point response to Comments and Suggestions for Authors

Comments 1: The study lacks power calculation. With 18-42% annual variations in case volume, was the sample size adequate to detect clinically meaningful survival differences (e.g., 5% absolute change in 1-year survival)?

Response 1: We thank the reviewer for raising this important point. A formal sample size calculation was performed prior to analysis using stratified random sampling, considering a 95% confidence level, maximum variability (p = 0.5), and 5% precision. Based on the total population of 539 patients with lung cancer recorded in the Almería province from 2019 to 2021, a minimum sample size of 224 was required. The final cohort included 530 patients, thus ensuring adequate statistical power. These details have now been added to the Methods section (2.4 – Page 6, Paragraph 3, Lines 238-244).

Comments 2: How were emergency vs. outpatient referrals triaged differently during pandemic peaks? This could explain the stable diagnostic intervals despite system stress.

Response 2: We appreciate this relevant observation. During the peaks of the COVID-19 pandemic, our institution implemented a prioritization protocol that ensured accelerated evaluation for patients referred from the emergency department. Outpatient referrals, particularly those from general practitioners, occasionally faced delays due to resource reallocation. This strategy likely contributed to the observed stability in diagnostic intervals across the study period. We have clarified this point in the Methods (section 2.4, Page 5, Paragraph 1, Lines 182-184) and expanded on it in the Discussion (Page 13, Paragraph 5, Lines 391-394).

Comments 3: For time-to-event analysis, why wasn't a competing risks model used given high mortality rates? COVID-19 deaths may have artificially inflated cancer survival estimates.

Response 3: We thank the reviewer for this important methodological observation. We acknowledge that a competing risks model could have provided a more refined analysis, especially given the context of the pandemic. However, cause-specific mortality data (e.g., death due to COVID-19 versus cancer) were not consistently available in our retrospective dataset. As a result, standard Cox proportional hazards models were used. We now explicitly state this limitation in the Discussion section and recommend future studies to include cause-of-death data to enable competing risk analysis (Page 14, Paragraph 10, Lines 443-445; Page 14, Paragraph 12, Lines 461-463).

Comments 4: The multivariate model doesn't adjust for comorbidities (e.g., COPD) that influence both treatment eligibility and survival. How might this affect the null finding for delay impacts?

Response 4: We agree that comorbidities, such as COPD or cardiovascular disease, can significantly affect both treatment selection and survival outcomes. Unfortunately, detailed comorbidity data were not consistently available across all patients in our retrospective dataset. While ECOG performance status was included in the multivariate model and may partially reflect underlying comorbidities, we recognize this as a limitation that could attenuate the observed associations between delay intervals and survival. This limitation is now explicitly addressed in the Discussion section (Page 14, Paragraph 10, Lines 445-447).

Comments 5: Molecular testing delays (11→15 days) are clinically significant for targeted therapy eligibility. Was there differential survival impact in biomarker-positive subgroups?

Response 5: We agree that molecular testing delays may be clinically relevant for patients eligible for targeted therapies. However, the relatively small number of patients with confirmed actionable mutations (e.g., EGFR, ALK) and the heterogeneity of available molecular data limited our ability to perform subgroup survival analysis with sufficient statistical power.

Comments 6: The 42% diagnostic rebound in 2021 exceeds most published recovery rates (typically 10-25%). Could this reflect pent-up demand from 2020, and were "catch-up" cases clinically distinct?

Response 6: We appreciate this important contextual observation. The 42% increase in lung cancer diagnoses observed in 2021 likely reflects both the resumption of diagnostic activities and pent-up demand from delayed or deferred cases in 2020. However, when comparing key clinical characteristics—such as ECOG performance status, stage at diagnosis, and treatment eligibility—we did not observe major differences between 2021 cases and those from other years. These data suggest that “catch-up” cases were not clinically distinct. We have added this interpretation to the Discussion section (Page 12, Paragraph 2, Lines 355-360).

Comments 7: How do you reconcile stable treatment intervals with documented pathology/molecular delays? Were treatment protocols modified (e.g., empiric chemo before biomarker results)?

Response 7: We thank the reviewer for this thoughtful question. Although pathology and molecular turnaround times were occasionally delayed, treatment protocols were adapted accordingly. In patients with aggressive disease, clinical judgment sometimes justified the initiation of empirical systemic therapy prior to receiving complete molecular results.

Comments 8: The improved survival despite pandemic stresses contradicts modeling studies predicting 5-15% excess mortality. Could lead-time bias from earlier diagnosis explain this?

Response 8: We agree that lead-time bias may partly explain the improved short-term survival observed during the pandemic. Earlier diagnosis, even without changes in disease biology or treatment efficacy, could artificially extend observed survival. While our multivariate model adjusted for stage and performance status, we now acknowledge this as a potential source of bias in the Discussion section. (Page 14, Paragraph 10, Lines 448-451)

Comments 9: Why did palliative radiotherapy decline sharply (12.3%→4.4%, p=0.0048)? This may reflect resource reallocation rather than clinical factors.

Response 9: We thank the reviewer for this important observation. The significant decline in the use of palliative radiotherapy likely reflects a strategic reallocation of resources toward curative-intent therapies during the most constrained phases of the pandemic. Notably, the proportion of patients receiving curative treatment increased from 69.83% in 2019 to 73.30% in 2021. This trend suggests that our institution prioritized potentially life-prolonging interventions over symptom-directed therapies. Furthermore, some patients may have declined hospital-based palliative treatments due to infection risk. We have clarified this in the Discussion (Page 13, Paragraph 5, Lines 408-411).

4. Response to Comments on the Quality of English Language

Point 1: Major Revision Recommendation

Response 1: Thank you for this important observation. We have thoroughly revised the manuscript to enhance the clarity, grammar, and scientific expression throughout the text. The language has been carefully edited by native English speakers with academic expertise in biomedical research. We believe the current version meets the standards of a high-impact scientific journal and is now more accessible to an international readership.

5. Additional clarifications

We confirm that all revisions have been implemented directly in the revised manuscript using track changes. All statistical analyses, figures, and tables have been reviewed for clarity and consistency. Where applicable, footnotes, legends, and methodological details were refined to meet the journal’s standards. We remain fully available to provide any further clarification or additional documentation if requested by the Editorial Office

Reviewer 2 Report

Comments and Suggestions for Authors

This review article submitted to Cancers J by MDPI, titled “Lung Cancer Under Siege: Timeliness, Treatment, and Survival Before and After the COVID-19 Pandemicby Blanco-Villar et al., 2025

  • In the current research the authors addressed the impact of the COVID19 pandemic on the diagnosis, treatment timing, and survival of lung cancer patients at a large public hospital in Spain.

Authors compared three consecutive years—before, during, and after the peak pandemic period—to see how care patterns evolved, where despite major disruptions in healthcare, lung cancer care was largely preserved. Diagnosis and treatment times remained stable, and short-term survival even improved during the pandemic. These findings showed that a strong, adaptive health system can maintain high-quality cancer care during times of crisis. The authors claim that the current results offer valuable insight into how hospitals can prepare for future challenges without compromising cancer outcomes.

  • The topic is original and relevant to the field and interesting
  • The abstract is structured, with details,
  • Key words are enough,
  • In the Introduction, the second paragraph is without ref. and this is not correct
  • The aim is well written, but per the aim addresses only Spain this should be reflected in the title, abstract conclusion and the manuscript conclusion
  • List of abbreviations is provided

Minor corrections:

  1. The introduction is obscure and needs clarification and more details,
  2. Several sentences are without ref.
  3. Please split long sentences and add a ref. for each single sentence or each single info.

  1. The “strength(s)” of the study to be mentioned
  2. Limitations to be added
  3. Add the future directions

Major corrections:

  1. The methods part, Figure 1 flow chart is confusing and needs to be redrawn.
  2. The inclusion and exclusion criteria are not clear and needs to be separated as a subheading,
  3. The expected out comes to be written
  4. Define the comparator
  5. A flow chart to be added for part 2.3 for data collection
  6. Table 1,3,4,5 p value for comparison between which years
  7. Below table 1,3,4,5,6 add the statistical method and program used and the level of significance
  8. In Table 1,3, 4,5,6 better to use NS for p values that are nonsignificant better than the value and only leave the significant values
  9. Table 2 is obscure and not understandable?

Author Response

For research article

Response to Reviewer 2 Comments

1. Summary

We sincerely thank Reviewer for their careful evaluation and constructive feedback on our manuscript. We appreciate the recognition of the originality and relevance of our work, as well as the detailed comments and suggestions provided. In response, we have revised the manuscript thoroughly, addressing both minor and major concerns. Specifically, we have improved the clarity and referencing of the Introduction, clarified the methodology and flowcharts, reorganized the presentation of the results, and strengthened the conclusions. All modifications have been implemented using track changes in the revised version of the manuscript. Below, we provide a detailed point-by-point response to each comment.

2. Questions for General Evaluation

Reviewer’s Evaluation

Response and Revisions

Does the introduction provide sufficient background and include all relevant references?

Must be improved

The Introduction has been revised for clarity and depth. Several references were added, and long sentences were split or restructured. Now it provides a more comprehensive context for the study.

Are all the cited references relevant to the research?

Must be improved

References have been updated and expanded to ensure relevance and scientific validity. Unreferenced statements were corrected.

Is the research design appropriate?

Must be improved

The design was clarified, including the comparator definition, statistical approach, inclusion criteria, and cohort structure. These are now better reflected in the Methods section.

Are the methods adequately described?

Must be improved

The Methods section was revised in detail: a new flowchart was added, timelines specified, and statistical methods annotated below all relevant tables.

Are the results clearly presented?

Must be improved

Tables were simplified and clarified, "NS" was used where appropriate, and table notes now indicate comparison years and methodology. Table 2 was reformatted.

Are the conclusions supported by the results?

Must be improved

The Conclusions now explicitly reflect the results and limitations and acknowledge that findings apply primarily to the Spanish healthcare context.

3. Point-by-point response to Comments and Suggestions for Authors

Comments 1:

  • The topic is original and relevant to the field and interesting
  • The abstract is structured, with details,
  • Key words are enough,
  • In the Introduction, the second paragraph is without ref. and this is not correct
  • The aim is well written, but per the aim addresses only Spain this should be reflected in the title, abstract conclusion and the manuscript conclusion
  • List of abbreviations is provided

Response 1: We thank the Reviewer for highlighting the strengths of our manuscript, including the originality and relevance of the topic, the structured and detailed abstract, the adequacy of the keywords, and the inclusion of the list of abbreviations. We also appreciate the reviewer’s constructive suggestions. In response to the suggestions:

·       We revised the second paragraph of the Introduction and added appropriate references (page 2, lines 59–69).

·       The title has been modified to specify the national scope: "Lung Cancer Under Siege in Spain...".

·       The final sentence of the abstract (page 2, Lines 47-48) and the Conclusions section (page 15, Lines 484-485) now explicitly refer to the Spanish healthcare system.

Comments 2:

Minor corrections:

  1. The introduction is obscure and needs clarification and more details,
  2. Several sentences are without ref.
  3. Please split long sentences and add a ref. for each single sentence or each single info.

  1. The “strength(s)” of the study to be mentioned
  2. Limitations to be added
  3. Add the future directions

Response 2: We appreciate these observations and have acted accordingly:

·       The Introduction was revised for clarity and readability (Page 2, Paragraph 2-4, Lines 59-81, 109-112), with long sentences split and additional references included.

·       A structured limitations paragraph was added to the Discussion (page 14, paragraph 10, Lines 438-451), explicitly addressing the retrospective design, lack of comorbidity adjustment, absence of cause-specific mortality data, and potential lead-time bias.

·       Additionally, the strengths of the study were outlined in the following paragraph (Discussion, page 14, paragraph 11, Lines 452-4560), including the robust sample size, three-year consecutive design, real-world data from a tertiary Spanish center, and the integration of diagnostic, treatment, and survival endpoints.

·       Future directions are now explicitly described in the last paragraph of the Discussion (page 14, paragraph 12, Lines 461-468).

Comments 1: Major corrections:

  1. The methods part, Figure 1 flow chart is confusing and needs to be redrawn.
  2. The inclusion and exclusion criteria are not clear and needs to be separated as a subheading,
  3. The expected out comes to be written
  4. Define the comparator
  5. A flow chart to be added for part 2.3 for data collection
  6. Table 1,3,4,5 p value for comparison between which years
  7. Below table 1,3,4,5,6 add the statistical method and program used and the level of significance
  8. In Table 1,3, 4,5,6 better to use NS for p values that are nonsignificant better than the value and only leave the significant values
  9. Table 2 is obscure and not understandable?

Response 1: We thank the reviewer for this detailed feedback. In response:

1.     Figure 1 has been redrawn using a clearer design (page 4).

2.     Inclusion and Exclusion Criteria are now in a separate subsection (2.2, page 4, Lines 144-154).

3.     The expected outcomes were added to the end of the Introduction (page 3, final paragraph, Lines 109-112).

4.     The comparator group was defined in the Study Design section (page 3, Paragraph 2, Lines 128-129).

5.     A new flowchart (Figure 2) was created to illustrate the data collection process (page 5).

6.     We have clarified in the methods and table legends that all p-values refer to comparisons across the three annual cohorts (2019, 2020, and 2021), using chi-square or Kruskal–Wallis tests as appropriate.

7.     The statistical method, software (R v4.1.2 and SPSS v26), and significance threshold (p < 0.05) have been explicitly included in the legends of Tables 1, 3, 4, 5, and 6.

8.     Following the reviewer’s suggestion, all non-significant p-values have been replaced with “NS”, while significant p-values (p < 0.05) remain in numeric form. This enhances clarity and visual emphasis.

9.     Table 2 has been reformatted for clarity and now includes a simplified structure and explanatory note: “2021 and 2021 are compared to the inmediately predecing year. All figures are descriptive” to indicate its purpose and comparator more clearly.

We believe these modifications address all concerns and improve both the readability and methodological transparency of the data presentation.

4. Response to Comments on the Quality of English Language

Point 1: The English is fine and does not require any improvement.

Response 1: We thank the reviewer for their positive feedback regarding the language quality.

5. Additional clarifications

We confirm that all changes in the revised manuscript have been marked using the track changes function. Additionally, all figures and tables have been reviewed to ensure clarity, coherence, and adherence to the journal’s formatting standards. Should any further modifications be necessary, we remain fully available to address them promptly. We appreciate the reviewer’s and editorial team’s efforts and thoughtful feedback.

Reviewer 3 Report

Comments and Suggestions for Authors

Summary: This manuscript presents a three-year cohort study from a Spanish tertiary hospital and assessed an in-depth analysis of the COVID-10 pandemic impact on lung cancer care, including diagnostic volumes, treatment patterns, wait times, and survival outcomes.

Major comments: 1. Authors need to comment on the power analysis for their study.

2. I have not found any information regarding the impact of patient source in this study. Were they all get diagnosed in the same hospital/clinic? 

3. I am curious to know if the same conclusion can be reached with other types of malignancies.

Author Response

For research article

Response to Reviewer 3 Comments

1. Summary

We thank the reviewer for their careful assessment and thoughtful feedback, which helped us improve the clarity, methodological rigor, and clinical relevance of the manuscript. Please find below our detailed responses. All changes have been incorporated into the revised manuscript and are highlighted in track changes.

2. Questions for General Evaluation

Reviewer’s Evaluation

Response and Revisions

Does the introduction provide sufficient background and include all relevant references?

Yes

Not applicable

Are all the cited references relevant to the research?

Can be improved

Clarified in Methods section (2.1)

Is the research design appropriate?

Can be improved

Revised power analysis and patient source added in section 2.1

Are the methods adequately described?

Yes

Not applicable

Are the results clearly presented?

Yes

Not applicable

Are the conclusions supported by the results?

Yes

Not applicable

3. Point-by-point response to Comments and Suggestions for Authors

Comments 1: Authors need to comment on the power analysis for their study.

Response 1: Thank you for pointing this out. We agree with this comment and have now included a detailed description of the sample size calculation in the Statistical Analysis subsection (page 7, paragraph 3, lines 238–244).

“A sample size calculation was performed using stratified random sampling and Epidat 4.2 software… [full text updated as shown in the revised manuscript].”

Comments 2: I have not found any information regarding the impact of patient source in this study. Were they all get diagnosed in the same hospital/clinic? 

Response 2: Thank you for this relevant question. We have clarified this issue in the revised manuscript. As explained in the final paragraph of the Introduction (page 3, paragraph 4, lines 109-112), our study hypothesized that the pandemic would negatively impact lung cancer diagnosis and treatment pathways. This hypothesis was tested across three consecutive annual cohorts (pre-pandemic, pandemic year one, and pandemic year two), regardless of the diagnostic hospital. Furthermore, we added a clarifying paragraph to the Methods (page 4, paragraph 1, lines 115-123), explaining that although some patients were initially diagnosed at regional centers within the province, all individuals included in the analysis were ultimately referred to the reference center for treatment planning and delivery of their first oncologic intervention. This centralized treatment model ensured consistency in data collection and outcome measurement.

Comments 3: I am curious to know if the same conclusion can be reached with other types of malignancies.

Response 3: Thank you for this insightful comment. While our study focused exclusively on lung cancer, we acknowledge that pandemic-related disruptions likely affected other cancer types as well. However, lung cancer is particularly sensitive to delays due to its aggressive biology and rapid progression. For this reason, our findings cannot be directly extrapolated to other malignancies without further evidence. We have modified the final paragraph of the Discussion to reflect this consideration and encourage future studies exploring the impact of the pandemic on other tumor types. (Page 18, final paragraph of Discussion, Lines 461-464)

4. Response to Comments on the Quality of English Language

Point 1: The reviewer indicated the English is fine and does not require any improvement.

Response 1: We thank the reviewer for their positive feedback regarding the language quality.

5. Additional clarifications

We have reviewed the manuscript thoroughly to ensure full compliance with journal formatting requirements, particularly in reference style (first 10 authors followed by et al. where applicable). We have also updated the figure legends and tables to reflect statistical methods, software used, and significance levels as requested in prior comments. All changes are tracked and highlighted in the revised manuscript.

Round 2

Reviewer 1 Report

Comments and Suggestions for Authors

Accept in present form